# Toxicological Effects of Glufosinate-Ammonium-Containing Commercial Formulations on *Biomphalaria glabrata* in Aquatic Environments: A Multidimensional Study from Embryotoxicity to Histopathology

**DOI:** 10.3390/toxics13070528

**Published:** 2025-06-24

**Authors:** Yuncheng Qian, Jialu Xu, Yilu Feng, Ruiqi Weng, Keda Chen, Hezheng Zheng, Xianwei Li, Qingzhi Zhao, Xiaofen Zhang, Hongyu Li

**Affiliations:** Key Laboratory of Artificial Organs and Computational Medicine in Zhejiang Province, Shulan International Medical College, Zhejiang Shuren University, Hangzhou 310015, China; yunchengqian0831@gmail.com (Y.Q.); xujialuuuuu@gmail.com (J.X.); yilufeng1207@gmail.com (Y.F.); wengruiqi020609@gmail.com (R.W.); chenkd@zjsru.edu.cn (K.C.); zhenghezheng34@gmail.com (H.Z.); xianweili749@gmail.com (X.L.); qingzhizhao33@gmail.com (Q.Z.); jxxfzhang@126.com (X.Z.)

**Keywords:** glufosinate-ammonium, *Biomphalaria glabrata*, embryotoxicity, aquatic ecology, histopathology

## Abstract

Glufosinate-ammonium (GLA) is a broad-spectrum herbicide widely used for weed control. However, its potential toxic effects on non-target aquatic organisms, especially in freshwater ecosystems, are of growing concern. This study investigates the toxic effects of GLA on *Biomphalaria glabrata*, a freshwater snail highly sensitive to environmental pollutants and commonly used as a model organism in toxicological studies. Acute toxicity tests revealed that the 96-h LC50 of GLA for adult snails was 3.77 mg/L, indicating moderate toxicity, while the LC50 for embryos was 0.01576 mg/L, indicating extremely high toxicity. Chronic exposure experiments further showed that at high concentrations (0.5 mg/L), the shell diameter and body weight of the snails not only failed to increase but also decreased, and they ceased to lay eggs. Moreover, their hepatopancreas and gonads suffered significant damage. Even at an environmentally relevant concentration of 0.05 mg/L, the body length, body weight, and reproductive capacity of the snails were inhibited, and damage to the hepatopancreas and gonads was observed. These findings provide important data for assessing the potential risks of GLA to aquatic ecosystems and offer a scientific basis for formulating environmental protection policies and optimizing herbicide usage standards.

## 1. Introduction

In recent years, the extensive use of glyphosate has led to the development of resistance in many weeds, prompting the widespread application of glufosinate-ammonium (GLA) as an alternative herbicide. GLA is a broad-spectrum, non-selective contact herbicide that acts by inhibiting glutamine synthetase in plants, preventing the normal synthesis of glutamine and leading to toxic accumulation of ammonia, which disrupts the basic metabolic processes of plants and ultimately causes plant death [1]. However, recent studies have shown that the toxicity of GLA may not only stem from the inhibition of glutamine synthetase; the generation of reactive oxygen species (ROS) is also considered to play a significant role in its toxic effects. Lipid peroxidation damage caused by ROS can lead to cell membrane disruption and, ultimately, cell death [2]. Current research has primarily focused on the toxic effects of GLA on humans and vertebrates. For instance, in humans, the toxicity of GLA is mainly manifested in its ability to induce hyperammonemia. Since GLA irreversibly inhibits glutamine synthetase, leading to significant accumulation of ammonia in cells, hyperammonemia is widely regarded as the core mechanism of its toxicity [3]. Although it is not yet clear whether hyperammonemia leads to neurotoxicity, it is generally believed that GLA poisoning may cause severe neurological complications, such as impaired consciousness, epileptic seizures, or central apnea. However, recent studies have shown that the correlation between GLA and neurological complications may be relatively low [4,5]. Despite this, the potential impact of GLA on the reproductive system has been confirmed. In humans, concentrations of GLA as low as 1 nM can reduce the respiratory efficiency of sperm mitochondria [6]. Similarly, in male mice, GLA can induce changes in sperm H3 lysine 4 trimethylation and histone H3 lysine 27 acetylation, which may adversely affect subsequent embryonic development and the health of offspring [7]. In female mice, maternal exposure to GLA can lead to synaptic pathology and indirectly alter the expression of genes regulating synaptic development, thereby producing certain neurogenic changes [8].

As a non-selective herbicide, GLA can eliminate various types of weeds. In soil, GLA can be rapidly degraded by microorganisms, with no residual activity [9]. Studies have shown that its toxicity to the soil organism *Eisenia fetida* is relatively low (LC50 > 2000 mg/kg), but even under low-concentration exposure, GLA can significantly reduce the weight, cocoons, and larvae of earthworms [10]. However, due to its high water solubility (solubility of 1370 g/L), the photodegradation rate of GLA in aquatic environments is relatively slow (with photodegradation half-lives of 1155 and 866 h under 4500 and 8300 lx, respectively, and 462 h and 40 h under UV light at 365 nm and 254 nm, respectively) [11]. This characteristic facilitates the migration of GLA from agricultural soils to adjacent freshwater environments, potentially causing water pollution [12]. Relevant studies have detected GLA in the surface water, sediments, and organisms of fish, shrimp, and crab aquaculture ponds [13].

The toxic effects of GLA are particularly pronounced in aquatic organisms. In zebrafish, GLA can induce liver injury and inhibit the Nrf2 signaling pathway, thereby exacerbating hepatotoxicity under high-concentration exposure [14]. In the freshwater crayfish *Procambarus clarkii*, sub-acute exposure to GLA leads to tissue damage in the hepatopancreas and weakens antioxidant and nonspecific immune capabilities [15]; For *Xenopus laevis* embryo-larvae, GLA exhibits moderate toxicity with significant concentration dependence, and exposure can slow growth [16]. In addition, GLA may trigger excessive stimulation of the hypothalamic-pituitary-thyroid axis by upregulating the production of thyroid hormones or directly reducing or inhibiting circulating thyroid hormones, thereby affecting normal development [17]. Although some research has been conducted on aquatic animals, further in-depth exploration is still needed for studies on aquatic invertebrates.

The freshwater gastropod *Biomphalaria glabrata* is widely distributed in Latin America and the Caribbean and is an intermediate host of *Schistosoma mansoni*. Its ecological and medical significance has made it a research hotspot [18,19]. *B. glabrata* is a hermaphroditic species that can reproduce both through self-fertilization and cross-fertilization, and its genome has been fully sequenced [20]. As a highly sensitive organism to toxic substances [21], *B. glabrata* is considered an excellent model organism [22]. Currently, *B. glabrata* has been widely used in studies of toxicity, embryotoxicity, genotoxicity, and bioaccumulation [23,24,25]. Recent studies have shown that exposure of *B. glabrata* to substances such as iron oxide nanoparticles (IONPs) can lead to behavioral disorders and reproductive toxicity [26], and it has also been used to monitor the concentration of natural radionuclides in groundwater [27]. These studies fully demonstrate the importance of *B. glabrata* in ecotoxicological research.

Given the high sensitivity of *B. glabrata* to environmental pollutants, it has become an ideal model organism for studying the toxic effects of GLA. By observing the development of snail embryos exposed to different concentrations of GLA over 120 h, phenomena such as delayed embryonic development, deformities, and even death, as well as the effects on adults in the 96-h acute toxicity test, such as escape behavior, hemolymph vomiting, and even death, the LC50 was calculated. Subsequently, a 21-day chronic toxicity experiment was further conducted, focusing on the examination of changes in snail body length, weight, reproductive capacity, damage to the hepatopancreas and gonads, and variations in hemolymph density. This study aims to comprehensively understand the acute toxicity of GLA on *B. glabrata* adults and embryos and the chronic toxic effects on adults. The findings not only reveal the toxic effects of GLA on individuals and embryos of *B. glabrata* but also provide a theoretical basis for understanding the potential impacts of herbicide pollution on freshwater ecosystems.

## 2. Materials and Methods

### 2.1. Breeding and Embryo Collection of B. glabrata

Adult *B. glabrata* (approximately 10 mm in diameter) were used in the experiments. During the experiments, the snails were maintained in transparent polystyrene plastic boxes measuring 270 × 180 × 100 mm, with 10 snails per liter. The water used was artificial pond water that had been aerated and dechlorinated, formulated according to the artificial pond water provided by the Biomedical Research Institute (pH 7.0 ± 0.2) (www.afbr-bri.org, accessed on 4 August 2024). The snails were maintained under laboratory conditions in accordance with OECD Guideline No. 243 [28]. The breeding environment was maintained at a temperature of 25 ± 2 °C with a 12-h light/dark cycle. To reduce water evaporation, the top of the plastic box was covered with a transparent, colorless plastic plate, with gaps on the sides to ensure air circulation. Fresh lettuce leaves (1 g per 10 snails) were provided as food, and the water was changed every 3 days. A clean polystyrene foam board was placed inside the box as a suitable oviposition area, with sterilized river sand laid underneath to provide appropriate living conditions. All snail embryos used in the experiments were obtained from healthy individuals of uniform size.

### 2.2. Chemical Reagents

Glufosinate-ammonium (C_5_H_11_N_2_O_4_P, 10% aqueous solution) was purchased from Sangon Biotech Co., Ltd., Shanghai, China (CAS No.: A614229-0100). This reagent is of analytical grade and can be directly used for toxicological studies. The stock solution had a concentration of 100 mg/L and was used to prepare a series of GLA solutions at different concentrations by gradient dilution method, including 7 mg/L, 6 mg/L, 5 mg/L, 4 mg/L, 2 mg/L, 1 mg/L, 0.5 mg/L, 0.1 mg/L, 0.07 mg/L, 0.05 mg/L, 0.03 mg/L, 0.01 mg/L, and 0.005 mg/L. Given the high water solubility of GLA, all dilutions were carried out using ultrapure water, and no additional organic solvents were introduced during solution preparation. This design minimizes the risk of solvent-related effects and ensures that the observed biological responses primarily reflect the action of the test compound.

### 2.3. GLA Exposure

#### 2.3.1. Behavior, Survival, and Reproductive Changes in Adult *B. glabrata* Exposed to Different Concentrations of GLA

We initially conducted acute toxicity tests to determine the 96-h LC50 of GLA for adult *B. glabrata*. The GLA stock solution was diluted with ultrapure water and stored at 4 °C in the dark. Healthy adult *B. glabrata* (*n* = 25) were divided into five groups of five snails each and placed in sterile polystyrene cylindrical containers. The containers had a bottom diameter of 90 mm and a top diameter of 116 mm, and each was filled with 300 mL of GLA solution at different concentrations (3 mg/L, 4 mg/L, 5 mg/L, 6 mg/L, and 7 mg/L). Parallel experiments were conducted, with six replicates for each concentration. During the experiment, the number of dead snails was recorded every 24 h, and their behaviors were observed, including feeding (indicated by the amount of leftover lettuce leaves), egg-laying, defecation, escape behavior, shell retraction, distribution, and bleeding. The observations continued for 4 days. The criteria for determining the death of a mollusk were (i) absence of heartbeat, (ii) immobility, and (iii) release of hemolymph [26]. When a *B. glabrata* remained retracted for a long time, showed no movement within 10 s after being stimulated with tweezers, and exhibited no feeding, defecation, egg-laying, or movement, and its color changed from red to yellow, it was considered dead. We documented the abnormal behaviors through photography and counted the number of dead individuals. The data were analyzed using nonlinear regression, with the following formula:Y = 100/(1 + 10^[(LogLC50 − X) × HillSlope]^)

(X: logarithm of concentration; Y (survival rate): number of survivors/initial number; HillSlope: slope of the curve)

In the chronic toxicity experiment, seven concentrations of GLA were selected: 0 mg/L (control), 0.05 mg/L, 0.1 mg/L, 0.5 mg/L, 1 mg/L, 2 mg/L, and 4 mg/L. For each concentration group, five healthy adult *B. glabrata* individuals were randomly selected and used as independent biological replicates. Each snail was housed individually in a sterile polystyrene container (volume: 150 mL; top diameter: 62 mm; bottom diameter: 42 mm) filled with 50 mL of GLA solution at the designated concentration. All exposures were conducted in parallel under identical environmental conditions for a period of 21 days. The exposure solutions were renewed every 3 days with freshly prepared solutions of the same concentration. During the experiment, each snail’s body weight and shell size were measured every 3 days, and the number of eggs laid was recorded daily. On the final day, hemolymph was collected from all surviving individuals (including controls), and hemocyte concentrations were determined. The soft tissues were then dissected for histological analysis of the gonads and hepatopancreas to compare tissue alterations across different concentration groups.

#### 2.3.2. Embryonic Malformations, Delayed Development, and Mortality in *B. glabrata* Exposed to Different Concentrations of GLA

To assess the acute embryotoxic effects of GLA on *B. glabrata*, five concentrations were selected for exposure: 0 mg/L (control), 0.005 mg/L, 0.01 mg/L, 0.03 mg/L, and 0.05 mg/L. Healthy egg masses were collected within 12 h after spawning and randomly assigned to treatment groups. Exposures were conducted in sterile 48-well plates, with each well containing one egg mass and 1 mL of solution of the corresponding concentration. For each concentration group, five replicate wells were established, and each egg mass served as an independent biological replicate. The entire experiment was independently repeated three times to ensure reproducibility.

Embryo mortality was recorded every 24 h across the 120-h exposure period. In parallel, survival rate (SR), hatching rate (HR), and developmental status were evaluated daily using a Motic stereoscopic digital microscope. The embryonic developmental process was also documented at 24-h intervals using a high-definition digital imaging system (Moticam S5) until the conclusion of the experiment.

The embryonic development was divided into five stages: blastula, gastrula, trochophore, veliger, and Hipo-stage. During the observation, when embryonic development ceased or the formed snails no longer exhibited rotational movement, and phenomena such as pigment deposition or vesiculation occurred, the embryo was considered dead. The effects of GLA on embryonic development were analyzed by comparing the experimental and control groups, including delayed or accelerated development.

### 2.4. Changes in Hemocyte Numbers in B. glabrata Exposed to Different Concentrations of GLA

At the end of the chronic toxicity experiment, hematology analyses were conducted on the surviving *B. glabrata* from each experimental group. To ensure the accuracy and reliability of the hemolymph samples, the entire hemolymph collection process strictly followed aseptic procedures. Initially, the external surfaces of the snails were disinfected with 75% alcohol, followed by thorough rinsing with PBS buffer to remove any residual alcohol and other contaminants, thereby ensuring a sterile sampling environment. We utilized the cephalopedal hemolymph extraction method to collect hemolymph from *B. glabrata*. The collected hemolymph samples were immediately transferred to 8-strip PCR tubes. Hemocyte concentration was determined using the Trypan Blue exclusion method for viability and total cell count. Hemocytes were mixed with Trypan Blue dye in a 1:9 ratio, and a 20 µL mixture was prepared, from which 10 µL was loaded onto a hemocytometer for analysis. The hemocytometer was calibrated before each experiment according to the manufacturer’s instructions to ensure measurement accuracy. Each sample was measured in triplicate, and the average value was taken as the hemocyte concentration result, expressed in cells/mL.

### 2.5. Morphological Observations of B. glabrata Soft Tissues Exposed to Different Concentrations of GLA

To better observe the effects of chronic GLA exposure on the morphology of *B. glabrata* soft tissues, the snails were decollated using plastic forceps to remove the shell, leaving only the soft tissue. The soft tissues of surviving individuals from each concentration group were placed on clean white paper, and morphological changes were observed and photographed. Subsequently, the tissues were used for histological analysis.

### 2.6. Histopathological Analysis of Hepatopancreas and Gonads in B. glabrata Exposed to Different Concentrations of GLA

To observe potential structural changes, four snails from each experimental group exposed to 0.5 mg/L, 0.1 mg/L, 0.05 mg/L, and 0 mg/L GLA were randomly selected. Samples of the hepatopancreas, testes, and ovaries were collected, dehydrated through a gradient alcohol series, embedded in paraffin, and cut into 3 µm thick sections. The sections were then dewaxed, rehydrated, and stained with hematoxylin and eosin (H&E) for subsequent scanning and image analysis. The histological sections were digitally scanned at arbitrary magnifications (1–800×) using a bright-field slide scanner.

### 2.7. Data Analysis

All statistical analyses were performed using GraphPad Prism 8.0 software. The LC50 values in the acute toxicity and embryotoxicity experiments were calculated with nonlinear regression (curve fitting), using the regression model log (inhibitor) vs. normalized response variable slope, with a confidence interval set at 95%. In the chronic toxicity experiments, data were analyzed with an analysis of variance (ANOVA) to assess the effects of different concentrations and exposure times on growth rates and egg-laying quantities. Subsequently, independent samples t-tests were used to compare differences between the control and GLA-treated groups. All statistical tests were one-tailed, with a significance level set at α = 0.05; *p*-values < 0.05 were considered statistically significant. Unless otherwise specified, all data are presented as mean ± standard deviation (mean ± SD). All experimental graphs were plotted using GraphPad Prism 8.0.2. Prior to parametric testing, the normality and homogeneity of variance of the data were verified using the Shapiro–Wilk test and Levene’s test, respectively. If the data did not meet the requirements for parametric testing, the corresponding nonparametric methods were employed. Experimental data were only considered valid when the mortality rate in the control group was below 10% [29]. All data were analyzed using GraphPad Prism 8.0 software.

## 3. Results

### 3.1. Acute Toxicity of GLA to Adult B. glabrata

In this study, we assessed the acute toxicity of GLA to adult *B. glabrata*. Figure 1A shows the chemical structure of GLA, which contains a glutamate-like group that can effectively inhibit the activity of glutamine synthetase, thereby causing toxic accumulation of ammonia and metabolic disorders. Additionally, the chemical stability of GLA further enhances the persistence of its toxic effects. After exposure to high concentrations of GLA solution, *B. glabrata* exhibited a series of behavioral and physiological changes (Figure 1B), including shell retraction, escape behavior, mucus expulsion, hemolymph bleeding, color changes, and alterations in their position in the water. These reactions indicate that GLA induces stress responses in *B. glabrata* at high concentrations, likely as part of their self-protection mechanisms. To further evaluate the toxic effects of GLA, we analyzed the survival rate data of snails after 96 h and fitted a survival rate curve. The results show that the toxicity of GLA increases with concentration, and there is a negative correlation between concentration and survival rate (Figure 1C). Based on these data, the LC90, LC50, and LC10 of GLA in *B. glabrata* were 5.40 mg/L, 3.77 mg/L, and 2.63 mg/L, respectively (Figure 1D). According to the Globally Harmonized System of Classification and Labelling of Chemicals (GHS), the environmental risk of a test substance can be categorized into four classes based on its concentration and effect: (i) highly toxic: L(E)C50 < 1 mg/L; (ii) moderately toxic: L(E)C50 between 1 and 10 mg/L; (iii) slightly toxic: L(E)C50 between 10 and 100 mg/L; (iv) practically non-toxic: L(E)C50 > 100 mg/L [30]. Based on these classification criteria, GLA exhibits moderate toxicity to adult *B. glabrata*. In the acute toxicity experiments, we recorded the survival rate of *B. glabrata* every 24 h (Figure 1E). The results show that snails exposed to high concentrations (6 mg/L and 7 mg/L) almost completely died within the initial 48 h of the experiment, further confirming the significant acute toxic effects of GLA on *B. glabrata* at higher concentrations.

### 3.2. Chronic Exposure to GLA Results in Growth Inhibition, Morphological Changes, and Abnormal Hemolymph Parameters in B. glabrata

During the 21-day chronic GLA exposure experiment, snails in the higher concentration groups (4 mg/L, 2 mg/L, and 1 mg/L) died within the first 11 days of the experiment. In contrast, in the lower concentration groups (0.5 mg/L, 0.1 mg/L, and 0.05 mg/L), all individuals survived until the end of the experiment except for a few deaths in the 0.5 mg/L group (Figure 2A). We further statistically analyzed the surviving snails and found that compared to the negative control group, the body length (Figure 2B) and weight (Figure 2C) of *B. glabrata* in the dosed groups (0.05 mg/L, 0.1 mg/L, and 0.5 mg/L) showed significant growth inhibition. Specifically, the growth rate in the 0.1 mg/L group was significantly slowed (** *p* < 0.01, *n* = 5), while growth in the 0.5 mg/L group was almost completely arrested (*** *p* < 0.001, *n* = 5). Notably, the 0.5 mg/L group exhibited significant negative weight growth (*** *p* < 0.001, *n* = 5) (Figure 2C). The effects of growth inhibition, as indicated by both body length and weight, increased with the dosing concentration (Figure 2B,C). Subsequently, we collected hemolymph from each surviving snail. The hemolymph volume and hemocyte density in the negative control group were significantly higher than those in the dosed groups (Figure 2E,F). This result further reveals the significant inhibitory effects of GLA on the hemolymph system of *B. glabrata*. In terms of morphological observations, we decollated each snail and found significant changes in the soft tissue morphology of the dosed groups, especially the color changes in the hepatopancreas region. With increasing GLA exposure concentration, the hepatopancreas of *B. glabrata* changed from the dark brown of the negative control group to a lighter color (Figure 2D). To further investigate this phenomenon, we performed histopathological sectioning of the hepatopancreas tissue. The histological results showed that GLA exposure caused pathological changes in the hepatopancreas tissue. Compared to the neat and regular acinar structure, uniform digestive cell cytoplasm, and clear nuclei in the negative control group, the acinar structure in the 0.05 mg/L GLA exposure group showed slight disorder, with mild vacuolar degeneration of hepatocytes and slightly blurred acinar margins (Figure 3). In the 0.1 mg/L exposure group, the degree of acinar disorder increased, with most hepatocytes showing vacuolar degeneration and fat accumulation, and local cell necrosis and inflammatory infiltration also appeared, with unclear acinar margins (Figure 3). In the 0.5 mg/L exposure group, the acinar structure was severely damaged, with loss of cell polarity, widespread vacuolar degeneration, increased areas of cell necrosis, incomplete boundaries between acini, and inflammatory cell infiltration (Figure 3). These results indicate that chronic exposure to GLA not only significantly inhibits the growth of *B. glabrata* but also causes significant damage to its hemolymph and tissue structure, revealing the potential chronic toxic effects of GLA on mollusks.

### 3.3. GLA Exposure Significantly Affects Reproductive Output and Gonad Histopathology in B. glabrata

During the 21-day observation period, the negative control group of *B. glabrata* exhibited normal reproductive activity, with eggs being laid almost daily. In contrast, although *B. glabrata* exposed to 0.05 mg/L and 0.1 mg/L GLA still showed some reproductive activity with relatively stable egg-laying quantities, the overall number was lower than that of the negative control group (Figure 2G). Specifically, the egg-laying quantities of snails exposed to 0.05 mg/L and 0.1 mg/L GLA were significantly lower than those of the negative control group, with statistically significant differences (** *p* < 0.01, *n* = 5). For the high-concentration exposure group (0.5 mg/L), the number of eggs laid further decreased, with almost no reproductive activity observed, indicating that GLA has a highly significant inhibitory effect on their reproductive systems (**** *p* < 0.0001, *n* = 5, Figure 2G). To further investigate this phenomenon, we performed histopathological analysis on the gonads. The results showed that gonad lesions were dose-dependent on GLA exposure concentration. The gonad structure in the negative control group was normal, with tightly arranged oocytes in the ovaries, intact oocyte morphology, and evenly distributed chromatin; the testes contained a normal number of spermatocytes and sperm cells with stable spermatogenesis. In contrast, the gonads in the 0.05 mg/L GLA exposure group showed slight lesions, with a slight decrease in sperm formation rate and a small number of germ cell atrophy observed. In the 0.1 mg/L exposure group, gonad damage was more severe, with increased cell atrophy and necrosis, reduced germ cell numbers, and weakened spermatogenesis. In the 0.5 mg/L exposure group, significant pathological changes in the gonads were observed, with some cells showing atrophy and fatty degeneration, most cells having disordered structure and necrosis, and germ cells almost completely lost. The oocytes in the ovaries had abnormal morphology, uneven chromatin distribution, and some follicular cavities collapsed, with vacuoles visible in the interstitium; in the testicular tissue, the number of spermatocytes was significantly reduced, and the sperm formation rate was significantly decreased (Figure 4). These results indicate that GLA not only significantly inhibits the reproductive activity of *B. glabrata* but also damages its gonad structure and reproductive function in a dose-dependent manner. This finding provides important histopathological evidence for understanding the reproductive toxicity of GLA on mollusks.

### 3.4. Concentration-Dependent Toxic Effects of GLA on B. glabrata Embryonic Development

In addition to the significant impact on reproduction, we also observed that GLA had pronounced toxic effects on embryonic development. Except for the negative control group, none of the egg masses laid by the other treatment groups hatched successfully (see Figure 5). Based on this observation, we hypothesized that GLA has concentration-dependent toxic effects on *B. glabrata* embryos. To test this hypothesis, we conducted a 9-day acute embryonic toxicity experiment. The results showed that the survival rate of embryos decreased significantly with increasing GLA concentration, and the toxicity curve further quantified the intensity of toxicity (see Figure 6A,B). The LC50 value of GLA for 50% embryonic lethality was 0.01576 mg/L, and at a concentration of 0.05 mg/L, all embryos died, indicating that GLA has strong toxicity to *B. glabrata* embryos. By observing the structural changes and developmental arrest of embryos using a stereoscopic microscope, we further analyzed the distribution of developmental stages of embryos at different concentrations (see Figure 6C). According to the developmental stages of *B. glabrata* embryos (Figure 6D), embryos in the low-concentration group (0.005–0.01 mg/L) developed relatively normally, with most embryos progressing smoothly. In the medium-concentration group (0.03–0.05 mg/L), embryonic development was significantly delayed, with most embryos remaining in the trochophore larva or veliger larva stages, showing developmental stagnation. In the high-concentration group (0.07–0.5 mg/L), embryonic development was almost completely arrested, with most embryos remaining in the blastula or gastrula stages, especially in the 0.5 mg/L group, where the vast majority of embryos were arrested in the blastula stage and did not continue to develop. We also detailed the types of embryonic death (Figure 6E) and found that with increasing GLA concentration, the type of embryonic death shifted from specific to nonspecific. In the negative control group and low-concentration treatment groups (0.005–0.01 mg/L), embryonic death was mainly specific (i.e., vacuolization death), with a low proportion of nonspecific death. As the concentration increased, the specific death rate decreased significantly, while the nonspecific death rate increased, indicating that embryonic development was inhibited. In the 0.5 mg/L group, nonspecific death dominated, and specific death almost completely disappeared, further indicating that the inhibitory effect of high-concentration GLA on embryonic development reached an extreme, causing embryos to stagnate in the early developmental stages. Survival curve analysis showed that the higher the GLA concentration, the lower the embryonic survival rate, with the survival curve shifting to the left (see Figure 6F). In summary, GLA exhibited concentration-dependent toxic effects on *B. glabrata* embryos, significantly inhibiting their normal development and severely affecting their growth and survival. This study provides important evidence for further understanding the ecotoxicological effects of GLA on aquatic invertebrates.

## 4. Discussion

The impact of pesticides on aquatic organisms has increasingly attracted widespread attention [15]. Glufosinate-ammonium, as a common organophosphorus herbicide, can enter natural water bodies through groundwater infiltration and surface runoff [15,31]. Once in the water cycle, GLA enters aquatic ecosystems [12], where it acts as an alien substance that may adversely affect various aquatic organisms, thereby threatening biodiversity and disrupting ecological balance [12,32,33]. Although GLA is relatively safe when used at recommended doses, its usage has increased exponentially over the past few decades, potentially leading to significant impacts when used in excess. Prenatal exposure to GLA via drinking water in a murine model has been shown to induce neurobehavioral impairments in offspring and disrupt gut microbiota composition, indicating potential developmental and neurotoxic effects. [34].

We investigated the acute toxicity of GLA to adult and embryonic *B. glabrata*, with LC50 values of 3.77 mg/L and 0.0157 mg/L, respectively. For adult snails, GLA exhibits moderate toxicity, but its toxicity to embryos is extremely severe and should be classified as highly toxic [30]. Compared to adults, the LC50 for embryos is approximately 240 times lower, indicating that *B. glabrata* embryos are highly sensitive to GLA. Numerous studies on the concentration of GLA in surface and groundwater have shown that the maximum detected concentration of GLA in more than a dozen provinces in China is 13.15 µg/L [35]; in Italy, the maximum detected concentration is 0.72 µg/L [36]; and in various watersheds in North America, the maximum detected concentration is 15.5 µg/L [37]. GLA concentrations have been detected in water bodies in different regions. Therefore, GLA concentrations that may exist in aquatic environments are sufficient to cause fatal effects on *B. glabrata* embryos.

During sub-lethal dose chronic exposure experiments, we found that even at relatively low concentrations, the lifespan of *B. glabrata* was very short, with all individuals dying within 6 days, especially at concentrations of 4 mg/L and 2 mg/L. Even individuals that survived at lower exposure concentrations (such as 0.05 mg/L, 0.1 mg/L, and 0.5 mg/L) exhibited significant inhibition of shell diameter and body weight growth, with particularly noticeable negative growth at 0.5 mg/L. Notably, these concentrations are achievable in natural environments.

After completing the sub-lethal dose exposure experiments, we decollated the surviving snails and observed significant changes in the color and structure of their hepatopancreas. Further histopathological section analysis revealed that with increasing dosing concentration, the pathological changes in the hepatopancreas became more pronounced, eventually leading to disordered structure and severely affecting the survival of *B. glabrata*. A study has shown that GLA can cause liver injury in zebrafish, with changes in related enzymes, oxidative stress indicators, inflammatory factors, and apoptosis-related enzyme levels in liver tissue as the concentration increases [14]. Low concentrations of GLA activate the Nrf2 pathway, while high concentrations inhibit it, and this inhibitory effect is one of the important causes of liver injury [14]. Mixtures of pesticides containing GLA significantly damage the liver function of South American toad larvae by affecting detoxification capabilities and oxidative stress [38]. Moreover, GLA can increase the activity of glutathione S-transferase while decreasing the activity of acetylcholinesterase and carboxylesterase, thereby causing more complex toxic effects [39].

Throughout the chronic exposure experiment, we observed that the total hemolymph volume in all GLA-treated groups significantly decreased, and hemocyte counts also showed a downward trend. This indicates that GLA has a significant inhibitory effect on the hemocytes of *B. glabrata*. The immune system of *B. glabrata* primarily relies on hemocytes to maintain function, so the reduction in hemocyte volume and density will inevitably weaken their immune function, thereby affecting their survival capabilities in natural environments. Hemocyte numbers are a direct reflection of the immune state and are also one of the early warning indicators of infection in aquaculture [40,41]. Studies have shown that GLA has significant immunotoxicity to zebrafish embryos, manifested as reduced survival rates, morphological deformities, and decreased numbers of macrophages and neutrophils [42]. Additionally, GLA exposure can induce oxidative stress, increase the activity of antioxidant enzymes (such as CAT and SOD), and significantly regulate genes and pathways related to metabolism, redox status, and immunity, further affecting the regulation of inflammatory cytokines and chemokines [42]. Moreover, as a glutamine synthetase (GS) inhibitor, GLA not only shows potential in cancer treatment but also exhibits unique effects on macrophage reprogramming [43]. GLA can shift macrophages from the tumor-promoting M2 type to the anti-tumor M1 type, a process accompanied by enhanced glycolysis and significant inhibition of tumor invasion capabilities [43], providing a new perspective on the multiple roles of GLA in immune regulation.

In the sub-lethal concentration range, when concentrations were 0.05, 0.1, and 0.5 mg/L, we observed significant inhibition of the reproductive capacity of *B. glabrata*, which severely affected their egg-laying ability, with near-zero egg production at high concentrations. This effect may not only weaken the sustainability of the population but also indirectly affect other species by disrupting predator-prey relationships. We performed histological sections of the gonads based on this phenomenon and found gonadal tissue damage, germ cell atrophy, decreased sperm formation rates, and abnormal oocyte morphology in the ovaries. GLA has shown reproductive toxicity in several animal experiments. For example, in studies on male lizards, exposure to GLA-contaminated soil resulted in oxidative damage and lesions in the testes, along with changes in plasma sex hormone levels, indicating that GLA may affect male lizard reproductive functions by interfering with the endocrine system [44,45,46]. GLA has also been shown to negatively affect human sperm mitochondrial respiratory efficiency, potentially leading to decreased sperm motility and vitality, thereby reducing male fertility [6,44,47]. Additionally, GLA has exhibited clear reproductive toxicity in rats, causing vaginal bleeding, premature birth, embryonic loss before and after pregnancy, and fetal intrauterine death [44]. Another study found that GLA has reproductive toxicity in *Caenorhabditis elegans*, manifested as reduced numbers of eggs and offspring in the body and induced germ cell apoptosis [48].

In embryotoxicity tests, even at low exposure concentrations of 0.01 mg/L, GLA significantly affected embryos, showing concentration-dependent inhibitory effects. Under microscopic observation, embryos exhibited obvious malformations, delayed development, and death. The toxic effects of GLA significantly reduced the number of newborn *B. glabrata*, thereby affecting their population density and ecological balance. The selected exposure concentrations are consistent with actual levels that may be encountered in the environment, and the interactive effects of pollutants in real environments may exacerbate embryonic exposure risks far beyond experimental conditions. In aquatic environments, GLA has toxic effects on zebrafish embryos, manifested as malformations and death [35]. When zebrafish embryos were exposed to 1.6 µg/L GLA, the mortality rate of embryos significantly increased [49]. Another study found that although the body weight and organ coefficients of mother mice did not change significantly during the 8-day GLA treatment, the number of live pups per litter significantly decreased [34]. Additionally, GLA may induce human embryo implantation failure by inhibiting glutamine synthase, as glutamine plays a crucial role in the embryo implantation process [6].

GLA exhibits significant inhibitory effects on the growth, reproduction, and development of *B. glabrata*, both in adults and embryos. Moreover, studies have shown that GLA can accumulate in the liver, brain, and reproductive glands of *Procambarus clarkii*, zebrafish, earthworms, *Xenopus laevis* embryos, and lizards, inducing oxidative stress, disrupting the normal functions of hormones and enzymes, interfering with immune responses, and impairing fertility and mobility, further highlighting the potential risks of GLA exposure to organisms [50]. In addition, some studies suggest that GLA has neurotoxic effects on organisms. In patients with GLA poisoning, the concentration of ammonia in cerebrospinal fluid is significantly correlated with neurological complications, followed by the concentration of 1-methoxy-2-propanol [4]. Although ammonia levels in serum are rapidly cleared after ingestion and may have little impact on neurological complications, ammonia is still considered the main substance causing neurological complications in GLA poisoning [4]. However, another study found that in patients with GLA poisoning, serum ammonia levels did not significantly increase before the onset of neurological complications but increased significantly afterward [5]. Therefore, it cannot be assumed that hyperammonemia is the main cause of neurological toxicity in GLA poisoning, and further research is needed to clarify the exact mechanism.

## 5. Conclusions

In conclusion, our multidimensional investigation demonstrates that GLA exerts both acute and chronic toxic effects on *B. glabrata*, with severity closely related to exposure concentration and life stage. A clear dose-dependent trend was observed across multiple biological endpoints: higher GLA concentrations caused increased mortality, inhibited growth and reproduction, reduced hemocyte density, and induced significant histopathological alterations in hepatopancreas and gonads. Moreover, embryonic development was highly sensitive to even environmentally relevant concentrations of GLA, resulting in delayed development, structural malformations, and death.

The findings collectively indicate that GLA-induced toxicity may be mediated by oxidative stress, immune suppression, and endocrine disruption, leading to cascading effects on individual physiology and population sustainability. The differential sensitivity between embryos and adults also suggests that early life stages are particularly vulnerable to herbicide contamination. These cause-and-effect relationships underscore the ecological risks posed by GLA and emphasize the need for regulatory attention and environmental monitoring in freshwater systems. Future studies should aim to clarify the molecular mechanisms underlying these effects and assess the combined toxicity of GLA with other agricultural pollutants in complex aquatic environments.

## Figures and Tables

**Figure 1 toxics-13-00528-f001:**
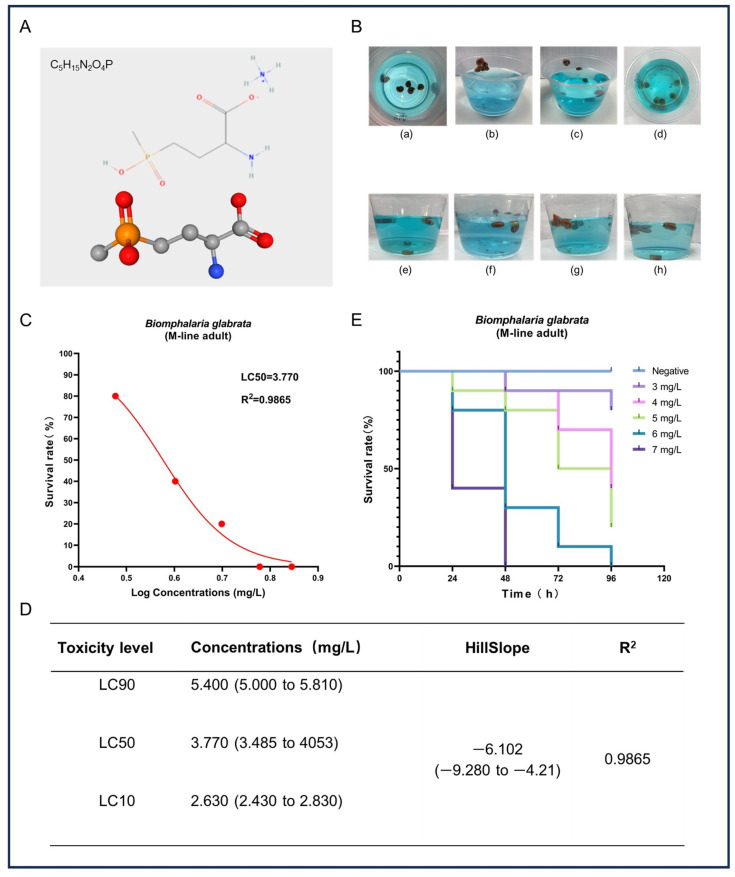
Acute toxicity effects of GLA on adult *B. glabrata*. (**A**) The upper part shows the chemical structure of GLA, and the lower part presents its ball-and-stick model. (**B**) Behavior and physiological changes of *B. glabrata* in water containing GLA: (**a**) When *B. glabrata* comes into contact with GLA solution (concentration range: 3–7 mg/L), it exhibits shell-retracting behavior; (**b**,**c**) In the low-concentration groups (3 mg/L, 4 mg/L) and medium-concentration groups (5 mg/L, 6 mg/L), snails are observed to escape, expel mucus, and change their positions in the water; (**d**) In the high-concentration group (7 mg/L) at the beginning of the experiment, a large number of individuals die; (**e**–**h**) show the distribution of snails in the low-concentration groups (3 mg/L, 4 mg/L) and medium-concentration groups (5 mg/L, 6 mg/L) in a 300 mL solution. (**C**) Non-linear regression analysis of dose-survival rate of adult *B. glabrata* exposed to GLA. (**D**) LC10, LC50, and LC90 values calculated based on the dose-survival rate fitting curve, 95% confidence intervals, curve slope, and R2 value. LC10: Concentration at which 10% of the experimental individuals die; LC50: Lethal concentration 50%; LC90: Lethal concentration 90%. (**E**) Survival curves of adult *B. glabrata* exposed to different concentrations of GLA, showing survival rates over time (96 h).

**Figure 2 toxics-13-00528-f002:**
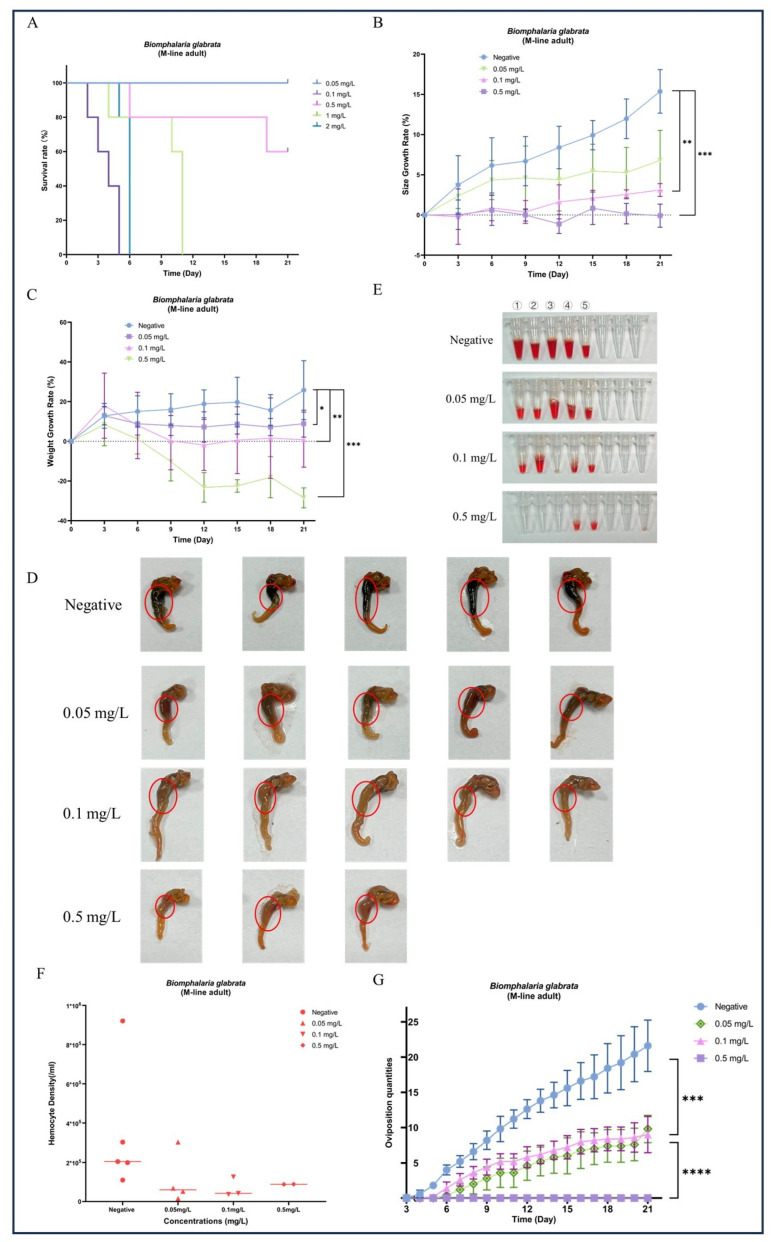
Effects of long-term exposure to GLA (21 days) on the growth, development, morphological changes, hemocyte density, and reproductive output of adult *B. glabrata*. (**A**) Survival rates of adult *B. glabrata* exposed to different concentrations of GLA (0.05–4 mg/L) over time. (**B**,**C**) Shell growth rate (%) and body weight gain (%) of *B. glabrata* exposed to different concentrations of GLA over time. Data are presented as mean ± SD (*n* = 5). Independent samples *t*-tests were used for comparisons, with significant differences between the control and treatment groups indicated by (* *p* < 0.05, ** *p* < 0.01, *** *p* < 0.001). (**D**) Morphological changes in *B. glabrata* after 21 days of exposure to different concentrations of GLA. The red circles highlight changes in the hepatopancreas of *B. glabrata* exposed to GLA. (**E**) Volume of hemolymph collected from *B. glabrata* after 21 days of exposure to different concentrations of GLA. Numbers 1–5 represent individual biological replicates, each corresponding to hemolymph collected from a separate snail within the same treatment group. (**F**) Hemocyte density in the hemolymph of *B. glabrata* after 21 days of exposure to different concentrations of GLA, as determined by hemocytometer counts. (**G**) Changes in the number of eggs laid by adult *B. glabrata* over time during 21 days of exposure to different concentrations of GLA (0.05 mg/L, 0.1 mg/L, and 0.5 mg/L). Data are presented as Mean ± SD, with significant differences indicated by asterisks (*** *p* < 0.01, **** *p* < 0.0001).

**Figure 3 toxics-13-00528-f003:**
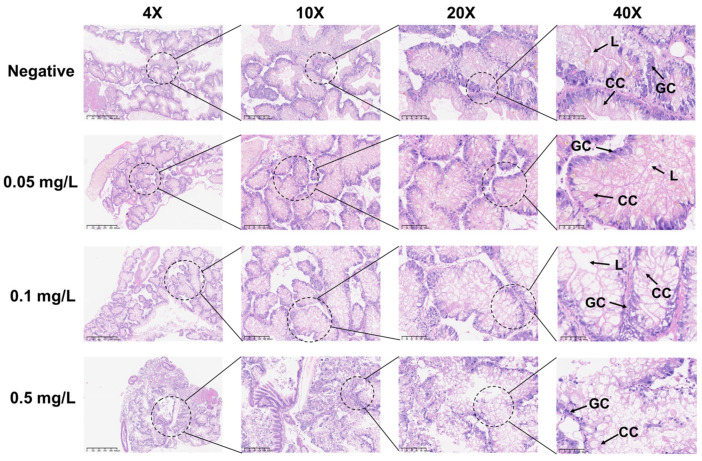
Histopathological changes in the hepatopancreas of adult *B. glabrata* exposed to GLA. Representative HE-stained sections of hepatopancreatic tissues after 21 days of exposure to different concentrations of GLA (from top to bottom: negative control, 0.05 mg/L, 0.1 mg/L, and 0.5 mg/L). The magnifications from left to right are 4× (scale bar = 625 μm), 10× (scale bar = 200 μm), 20× (scale bar = 100 μm), and 40× (scale bar = 50 μm). The figure labels the lumen (L), granular cell (GC), and cylindrical cell (CC).

**Figure 4 toxics-13-00528-f004:**
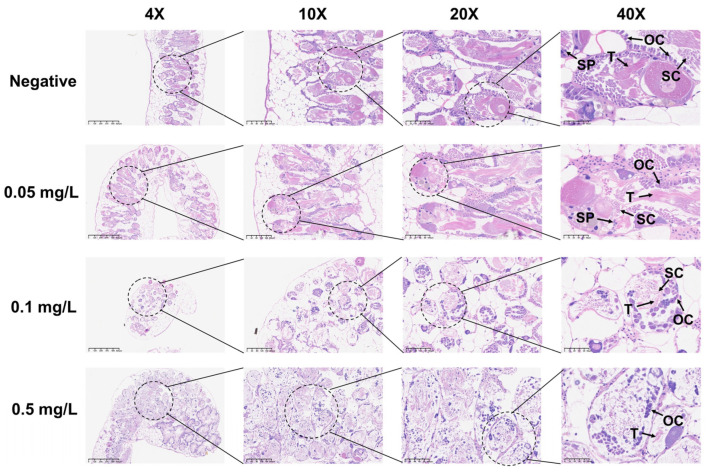
Histopathological changes in the reproductive glands of adult *B. glabrata* exposed to GLA. Representative HE-stained sections of reproductive gland tissues after 21 days of exposure to different concentrations of GLA (from top to bottom: negative control, 0.05 mg/L, 0.1 mg/L, and 0.5 mg/L). The magnifications from left to right are 4× (scale bar = 625 μm), 10× (scale bar = 200 μm), 20× (scale bar = 100 μm), and 40× (scale bar = 50 μm). The figure labels the ovary cell (OC), testis (T), spermatic cord (SC), and sperm (Sp).

**Figure 5 toxics-13-00528-f005:**
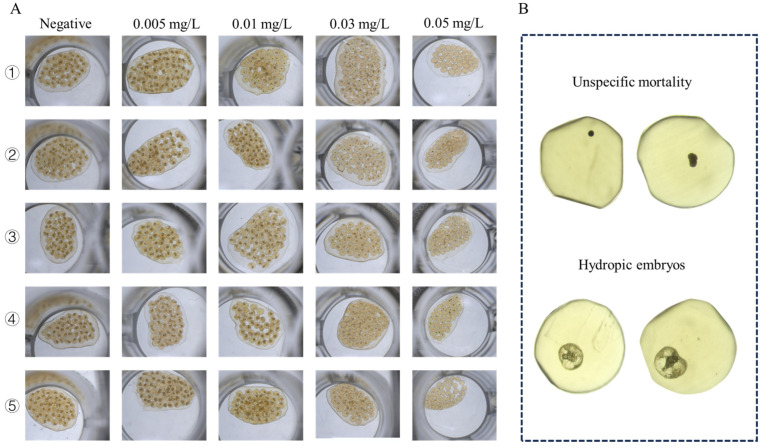
Microscopic images of *B. glabrata* embryos treated with different concentrations of GLA (0.005–0.5 mg/L). (**A**) Representative images of five embryos from each group after 120 h of treatment with different concentrations of GLA. (**B**) Two types of embryonic death observed: unspecific mortality (top two images) and hydropic embryos (bottom two images).

**Figure 6 toxics-13-00528-f006:**
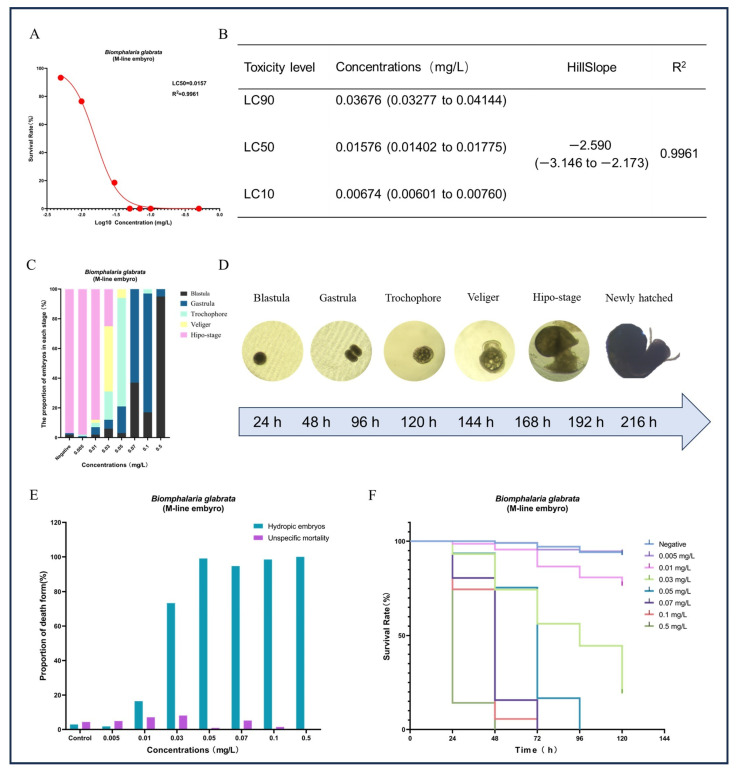
Effects of different concentrations of GLA on the survival, developmental process, and types of mortality in *B. glabrata* embryos. (**A**) Non-linear regression analysis of dose-survival rate for the 50% lethal concentration (LC50) of *B. glabrata* embryos exposed to different concentrations of GLA. (**B**) LC10, LC50, and LC90 values calculated based on the dose-survival rate fitting curve, along with 95% confidence intervals, curve slope, and R^2^ values. LC10: Concentration at which 10% of the experimental individuals die; LC50: Lethal concentration 50%; LC90: Lethal concentration 90%. (**C**) Proportion of embryos in different developmental stages (Blastula, Gastrula, Trochophore, Veliger, Hypo-stage) after 120 h of exposure to different concentrations of GLA. (**D**) Different stages of embryonic development in *B. glabrata*, including Blastula, Gastrula, Trochophore, Veliger, Hypo-stage, and New-hatched. Each stage is represented by images corresponding to different time intervals. (**E**) Proportion of different types of embryonic mortality under different concentrations of GLA. The figure compares two types of mortality: unspecific mortality and hydropic embryos. (**F**) Survival curves of embryos of *B. glabrata* exposed to different concentrations of GLA (0.005–0.5 mg/L) over time (0–120 h). Error bars indicate Mean ± SD from three independent experiments (*n* ≥ 5); The red dots in the figure represent the survival rate data for each treatment group after exposure.

## Data Availability

The data presented in this study are available on reasonable request from the corresponding author.

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
