# Peer review of "Toxicological Effects of Glufosinate-Ammonium-Containing Commercial Formulations on Biomphalaria glabrata in Aquatic Environments: A Multidimensional Study from Embryotoxicity to Histopathology"

_toxics, 2025, doi:10.3390/toxics13070528_

Round 1
Reviewer 1 Report
Comments and Suggestions for Authors
The article is good and comprehensive, a lot of work has been done, the area and topic is interesting, to improve the clarity of some parts, please see the suggestions in the report in the attachment. Manuscript is clear, relevant for the field and presented in a well-structured manner. References are relevant for the article and do not include an excessive number of self-citations.

Author Response
Response to Reviewer 1 Comments
Summary:
Thank you very much for taking the time to review this manuscript and for your comments on this manuscript. We value your positive comments and constructive suggestions. Please read the responses below one by one. All changes have been made and are reflected in the resubmitted document.
Comment 1:
“keywords recommendations: ,,Embryotoxicity, Histopathology" instead of ,,Toxicity: Growth and Reproduction”
Response 1:
Thank you for your suggestion. We have revised the keywords to “Embryotoxicity” and “Histopathology” to better reflect the study focus.
Comment 2:
“2.1.Ethical consideration - My opinion is that such declarations should be placed at the very end of the article, altogether with the acknowledgments.”
Response 2:
We appreciate your suggestion. We have therefore moved the ethics statement to the end of the manuscript, at lines 590-597.
Comment 3:
“general remark: Biomphalaria glabrata is sometimes not in the italic, please correct that (abstract, title 2.4.2.)”
Response 3:
Thank you for pointing this out. We have carefully reviewed the manuscript and corrected all instances of Biomphalaria glabrata to ensure consistent italic formatting.
Comment 4:
“Can you please add if you have been measuring replicates in your experiments?”
Response 4:
Thank you for your comment. We confirm that all experiments were conducted with replicates, and this information has been described in the manuscript. Specifically, in Section 2.3.1 (“Behavior, Survival, and Reproductive Changes in Adult Biomphalaria glabrata Exposed to Different Concentrations of GLA”), the number of replicates is indicated in Lines 142 and 174.
Comment 5:
“Can you please enlarge Fig.1., Fig.2. and Fig.3. for better visibility and clarity?”
Response 5:
Thank you for your valuable suggestion. In response, we have enlarged the corresponding figures and optimized their layout in the revised manuscript to enhance clarity and readability. High-resolution versions of these figures will also be provided as separate files during the submission process to facilitate detailed review if needed.
Comment 6:
“Can you please be more comprehensive in Conclusions, talk about general trends and cause-and-effect relationships, it would be possible because you have nice and comprehensive study.”
Response 6:
Thank you for your insightful comment. In response, we have revised the Conclusion section to provide a more comprehensive discussion of the observed trends and the potential cause-and-effect relationships derived from our experimental findings. The updated Conclusion can be found in Lines 571–587 of the revised manuscript.

Reviewer 2 Report
Comments and Suggestions for Authors
The Authors analyzed the acute toxic effects of Glufosinate-ammonium (GLA), a broad-spectrum herbicide, on Biomphalaria glabrata, a freshwater snail. The study is interesting, and the topic is relevant in the field of toxicological studies using freshwater model organisms.
Adults and embryos of Biomphalaria glabrata were exposed to environmentally significant concentrations. Behavior, Survival, and Reproductive Changes were analyzed on adults together with hemocyte count and histological analysis. Embryonic malformations, delayed development, and mortality were analyzed on embryos.
The conclusions are consistent with the main question posed, i.e., understand the acute toxicity of GLA on B. glabrata adults and embryos and the chronic toxic effects on adults.
The references are appropriate
In the following, some specific issue:
Lane 63: please write Eisenia fetida in Italics, ie., Eisenia fetida.
Line 275: please check the LC90, LC50, and LC10 values reported in the text with the same value reported in Figure 1 D.
Line 363, section 3.3, Figure 5A is introduced before figure 4, subsequently after figure 6 and figure 7 descriptions, on line 420-421, it is cited the developmental stages of B. glabrata embryos reported on Figure 5B, this is quite confusing. I suggest reconsidering figure 5 and 7 organizations, for example figure 5B could be moved in Figure 7, alternatively, change the description in the main test according to the figure’s organization.
Line 469-470: “GLA has not only been detected in the environment but also in the body fluids of pregnant women and newborns [34]” The reference cited concerns a study on a mouse model prenatally exposed to GLA through drinking water to evaluate neurobehavioral abnormalities in offspring and the role of gut microbiota in these diseases. So, change the sentence according to the reference cited.
Line 550: “GLA (GLA) has toxic effects…..” (GLA)” can be deleted?
Line 211-219. The Authors use the term blood and hemolymph interchangeably, in M&M, in the main text and in the caption of figure 2 but I think that this is potentially confusing. So, I suggest adding some explanation (in M&M, for example) or use only one term, hemolymph preferably.
Author Response
Response to Reviewer 2 Comments
Summary:
We sincerely thank you for your thorough review and thoughtful comments on our manuscript. In particular, we appreciate your positive comments on the design, clarity, and scientific relevance of this study. Your constructive suggestions will greatly help us improve the quality and rigor of our work. To this end, we have carefully addressed each point in detail, as outlined below. All corresponding revisions have been incorporated into the manuscript and are highlighted in yellow for your convenience.
Comment 1:
“Line 63: please write Eisenia fetida in Italics, ie., Eisenia fetida.”
Response 1:
Thank you for this observation. We have corrected the formatting of Eisenia fetida to ensure it is italicized.
Comment 2:
“Line 275: please check the LC90, LC50, and LC10 values reported in the text with the same value reported in Figure 1D.”
Response 2:
Thank you for pointing this out. We have carefully reviewed the values and corrected the discrepancies in the text to ensure consistency with those presented in Figure 1D. The revised values can be found in Line 265 of the updated manuscript.
Comment 3:
“Line 363, section 3.3, Figure 5A is introduced before figure 4, subsequently after figure 6 and figure 7 descriptions, on line 420-421, it is cited the developmental stages of B. glabrata embryos reported on Figure 5B, this is quite confusing. I suggest reconsidering figure 5 and 7 organizations, for example figure 5B could be moved in Figure 7, alternatively, change the description in the main text according to the figure’s organization.”
Response 3:
Thank you for your helpful suggestion. To improve the logical flow and clarity of figure presentation, we have reorganized the figures as follows: Figure 5A has been moved to Figure 2G, while Figures 5B and 5C have been relocated to Figure 7 as panels D and F, respectively. We have also adjusted the corresponding descriptions in the main text to match the new figure arrangement. These revisions can be found in Lines 426–428 of the updated manuscript.
Comment 4:
“Line 469-470: ‘GLA has not only been detected in the environment but also in the body fluids of pregnant women and newborns [34]’ The reference cited concerns a study on a mouse model prenatally exposed to GLA through drinking water to evaluate neurobehavioral abnormalities in offspring and the role of gut microbiota in these diseases. So, change the sentence according to the reference cited.”
Response 4:
Thank you for your careful reading and insightful comment. After re-examining the cited reference, we fully agree with your observation. The original sentence has been revised to more accurately reflect the content of the study as follows:“Prenatal exposure to GLA via drinking water in a murine model has been shown to induce neurobehavioral impairments in offspring and disrupt gut microbiota composition, indicating potential developmental and neurotoxic effects.”This revision can be found in Lines 464–467 of the updated manuscript.
Comment 5:
“Line 550: ‘GLA (GLA) has toxic effects…..’ (‘GLA’) can be deleted?”
Response 5:
Thank you for pointing out our oversight. We have deleted the redundant “(GLA)” in that sentence.
Comment 6:
“Line 211-219. The Authors use the term blood and hemolymph interchangeably, in M&M, in the main text and in the caption of figure 2 but I think that this is potentially confusing. So, I suggest adding some explanation (in M&M, for example) or use only one term, hemolymph preferably.”
Response 6:
Thank you for highlighting this inconsistency. We agree that “hemolymph” is the more accurate term for mollusks and have replaced “blood” with “hemolymph” throughout the manuscript, including the Materials and Methods section and figure captions.
